

# Focused ultrasound-induced cell apoptosis for the treatment of tumours

Na Wang[1,2,*], Li Luo[2,*], Xinzhi Xu[2], Hang Zhou[2] and Fang Li[2]

[1] Chongqing University, School of Medicine, Chongqing, China
[2] Chongqing University Cancer Hospital, Ultrasound Department, Chongqing, China
[*] These authors contributed equally to this work.

## ABSTRACT

Cancer is a serious public health problem worldwide. Traditional treatments, such as surgery, radiotherapy, chemotherapy, and immunotherapy, do not always yield satisfactory results; therefore, an efficient treatment for tumours is urgently needed. As a convenient and minimally invasive modality, focused ultrasound (FUS) has been used not only as a diagnostic tool but also as a therapeutic tool in an increasing number of studies. FUS can help treat malignant tumours by inducing apoptosis. This review describes the three apoptotic pathways, apoptotic cell clearance, and how FUS affects these three apoptotic pathways. This review also discusses the role of thermal and cavitation effects on apoptosis, including caspase activity, mitochondrial dysfunction, and $Ca^{2+}$ elease. Finally, this article reviews various aspects of FUS combination therapy, including sensitization by radiotherapy and chemotherapy, gene expression upregulation, and the introduction of therapeutic gases, to provide new ideas for clinical tumour therapy.

## INTRODUCTION

According to the latest global cancer burden data from the International Agency for Research on Cancer (IARC), the incidence of cancer has reached 20 million cases, with nearly 9.7 million deaths (*Bray et al., 2024*). The classical cancer treatment modalities include surgery, radiation therapy, chemotherapy, and immunotherapy. With the available treatment modalities and differences in cancer types and regions, the five-year survival rates of patients with various cancers significantly differ (*Bray et al., 2024*). Cancer has become one of the most important global public health issues. Thus, research on cancer treatment mobility is still of utmost importance and urgency. Guided by 2D greyscale ultrasound images and magnetic resonance (MR) (*Lamsam et al., 2018*; *Brighi et al., 2020*), focused ultrasound (FUS) is a noninvasive approach that utilizes external ultrasound waves to accurately target and concentrate regions. Apart from inducing coagulative necrosis in tumours through thermal ablation, FUS can also trigger tumour cell apoptosis through mechanical effects, thermal effects, and nonthermal and nonmechanical effects (*Wu, 2014*; *Wang et al., 2021b*). Apoptosis is an inherent surveillance and regulatory mechanism that enables the timely elimination of nonfunctional, harmful, and aberrant cells (*Newton et al.,*

Corresponding author
Fang Li, lifang0703@cqu.edu.cn

*2024*). Rewiring cancer drivers to activate apoptosis is also one of the tumour treatment modalities currently being pursued by many researchers, including strategies focusing on cell apoptosis to enhance the sensitivity and specificity of tumour therapy (*Gourisankar et al., 2023*). Notably, FUS had no apparent cumulative effect on the passage of FUS energy. Because of biochemical signalling, metabolism, and anchorage, malignant cells also differ from normal cells in terms of mechanical properties (*Kim & Hyun, 2023*). Malignant cells are more susceptible to FUS stimulation. Accordingly, in the context of cancer, FUS can induce the apoptosis of malignant cells and preserve the surrounding healthy cells as much as possible.

Investigating the mechanisms of FUS-induced apoptosis involves accurately evaluating the effective treatment area, optimizing the acoustic power, reducing the risk of complications, and exploring the possibility of FUS combination therapy. This review describes the main pathways of tumour cell apoptosis, the relevant mechanisms of FUS-induced tumour cell apoptosis, and current research on FUS application in tumour therapy. This review might provide basic knowledge for general readers as well as cancer researchers, especially those who are dedicated to tumour research and developing new methods of cancer treatment, which may provide new insights to improve the clinical diagnosis and treatment of tumours.

## SURVEY METHODOLOGY

Literature searches of the review for relevant studies were conducted in PubMed, Web of Science, and the China National Knowledge Infrastructure (CNIK). The keywords used were as follows: ultrasound, apoptosis, apoptosis pathways, death receptor pathway, endoplasmic reticulum stress signalling pathway, mitochondrial apoptosis pathway, and so on. Other keywords are shown in the appendix. We limited our literature search to the last 10 years, but we also considered other references in the cited literature to ensure that the literature was fully covered and screened. We screened a total of 397 articles, 128 of which were newer, of better quality, and more relevant to our topic.

### Tumour-related apoptosis pathways

Apoptosis is a type of orderly cell death controlled by genes to maintain the stability of the internal environment of the body. It involves the activation, expression, and regulation of a series of genes (*Newton et al., 2024*). Apoptosis is characterized by distinct histomorphological examinations: loss of cell membrane integrity, breakdown of cellular structures, aggregation of nucleoli and cytoplasm. Apoptosis manifests as cell shrinkage, the appearance of large transparent vacuoles in cytoplasm, chromatin condensation, orderly fragmentation of nuclear DNA, and ultimately the formation of apoptotic bodies (*Zhong et al., 2019*; *Cao et al., 2021*; *Xu et al., 2021*; *Dejas et al., 2023*). *Hanahan & Weinberg (2011)* proposed that the complexity of tumour diseases lies in the maintenance of proliferation signals, evasion of growth inhibitory factors, resistance to apoptosis, and immortality of cell replication. The doubling time of the tumour volume in malignant diseases is less than the doubling time of the tumour cell number (*Morana, Wood & Gregory, 2022*). Apoptosis is inhibited in the tumour environment to a certain extent. Tumour-related

apoptotic pathways include the death receptor pathway, the mitochondrial pathway, and the endoplasmic reticulum stress (ERS) pathway.

Due to the noncumulative effect of FUS and the fact that malignant cells are more sensitive to ultrasound waves, FUS induces malignant cell apoptosis but protects normal cells from it. The effective removal of cancer cells by programmed cell death is a feasible approach for clinical cancer treatment. Depending on the intensity of the ultrasound energy, FUS can be divided into high-intensity focused ultrasound (HIFU) and low-intensity focused ultrasound (LIFU). HIFU (greater than 200 W/cm$^2$) refers to the use of a special ultrasound transmitter to focus sound waves towards the tumour site, converting them into heat energy and causing the temperature of the tumour treatment site to reach 65–100 °C in approximately 0.25 s, resulting in degeneration and necrosis of malignant cells (*Jin, Zhao & Huang, 2023*). LIFU (less than 100 W/cm$^2$) also employs a focused emission sound field, with the sound beam converging towards the acoustic axis. It exhibits good focusing, penetration, and anti-attenuation properties but lacks the destructive properties of thermal ablation (*Zhong et al., 2023*). HIFU and LIFU can cause apoptosis through different pathways and different proteins. This part reviews three classical apoptotic pathways and describes how HIFU and LIFU affect these pathways.

### Death receptor pathway

A specific molecular death receptor on the surfaces of cells can bind to a specific molecule death ligand, and the resulting complex participates in activation of the caspase enzyme and the subsequent apoptotic reaction, which is the death receptor pathway of apoptosis (*Green, 2022*). Death receptors mainly include tumour necrosis factor receptor-1 (TNFR1), CD95 (also called Fas and APO-1), TRAIL receptors (also called DR4), TRAIL receptor-2 (also called DR5), death receptor 3 (DR3) and death receptor 6 (DR6). The main death ligands include TNF, CD95 ligand (CD95-L or Fas-L), and TRAIL (a TNF-related apoptosis-inducing ligand, also called APO-2L) (*Annibaldi & Walczak, 2020*).

CD95, as the most typical death receptor, can be used to elucidate the detailed process of apoptosis. CD95 is a trimer structure on the cell surface. When bound by the ligand CD95-L, its conformation changes, and the DD (death domain) is exposed to the cell. The DD then interacts with Fas-associated death domain protein (FADD) *via* DD-DD interactions, facilitating exposure of death effector domain (DED) in FADD. Consequently, the DED on the FADD can recruit caspase-8 and result in the oligomerization of caspase-8 through the DED-DED interaction. This complex, comprising ligated CD95, FADD, and caspase-8, is called DISC (DISC) (*Xue et al., 2023*; *Wang et al., 2023b*; *Ranjan & Pathak, 2024*). Activated caspase-8 initiates apoptosis directly by cleaving and activating the executioner caspase (−3, −6, and −7). Alternatively, it triggers the mitochondrial apoptosis pathway by cleaving Bid. The mitochondrial apoptotic pathway is described in the following sections and is reiterated here (*Kantari & Walczak, 2011*; *Kaufmann, Strasser & Jost, 2012*; *Lavrik & Krammer, 2012*; *Han et al., 2023*; *Mahadevan et al., 2023*; *Sahoo et al., 2023*; *Haymour et al., 2023*; *Ma et al., 2024*).

The mechanism of apoptosis induced by TRAIL receptors appears similar to that induced by CD95. The ligation of TRAIL receptors results in the recruitment of the FADD to DDs
in the intracellular region. Then, the FADD binds to and dimerizes caspase-8, thereby activating it *Yagolovich, Gasparian & Dolgikh (2023)* and *Yuan & Ofengeim (2023)*.

However, apoptotic signalling by TNFR1 is complex. As an extracellular apoptotic signal, TNF activates its cell surface receptor TNFR1 and forms a multicomponent protein complex (complex I) on the cytosolic side of the plasma membrane; this complex consists of adaptor proteins TRADD, TRAF2, receptor-interacting protein kinase 1 (RIPK1), and a pair of functionally redundant ubiquitin ligases with caspase-inhibiting activity (cIAP1 and cIAP2). Normally, RIPK1 is polyubiquitinated, and complex I activates the nuclear factor κB (NF-κB) signalling pathway, causing the transcription of various cytokines and antiapoptotic proteins and contributing to inflammation maintenance and cell death prevention. When the NF-κB pathway is inhibited, RIPK1 is phosphorylated, and the translation of antiapoptotic proteins is suppressed. Complex I dissociates from the plasma membrane and recruits the FADD and caspase-8 to generate a cytosolic caspase-8-activating complex (complex II), thereby triggering apoptosis (*Siegmund, Zaitseva & Wajant, 2023*; *Ai et al., 2024*) (Fig. 1).

### Mitochondrial apoptosis pathway

When cells are subjected to severe stress (such as growth factor withdrawal, extensive DNA damage, endoplasmic reticulum stress, cell hypoxia, radiation, and physical damage), the mitochondrial apoptosis pathway can be activated and induce apoptosis (*Vringer & Tait, 2023*). Bcl-2 family proteins are the most important regulators of the mitochondrial apoptosis pathway and can regulate the mitochondrial membrane potential and control the permeability of the mitochondrial outer membrane (MOMP). Upon proapoptotic stress, two proapoptotic proteins, Bax and Bak, are activated by BH3 only and relocalize to the surface of the mitochondrial outer membrane. Then, activated Bax and Bak induce MOMP, causing the release of cytochrome c. Cytochrome c binds to apoptosis protease activating factor 1 (APAF1), inducing the oligomerization of APAF1 into a heptameric structure called the apoptosome, which subsequently activates initiator caspase-9. Active caspase-9 then cleaves and activates effector caspases (caspase-3, caspase-6 and caspase-7), ultimately leading to apoptosis (*Flores-Romero, Dadsena & García-Sáez, 2023*; *Nguyen et al., 2023*; *Xie et al., 2023*; *Harrington et al., 2023*) (Fig. 2).

### Endoplasmic reticulum stress signalling pathway

As a complex and dynamic organelle, the endoplasmic reticulum (ER) is essential for normal cellular pathways, including $Ca^{2+}$ storage, lipid metabolism, and protein production (*Li et al., 2024*). When entering a stress state, the ER activates the unfolded protein response (UPR) (*Abbonante et al., 2023*). The UPR is a highly conserved mechanism in metazoans and consists of three ER-associated pathways that initiate adaptive transcriptional programs within the nucleus: PKR-like ER-resident kinase (PERK), activated transcription factor 6 (ATF6), and inositol-requiring enzyme 1 (IRE1) (*Jeong et al., 2023*). By sensing the accumulation of unfolded proteins or lipid bilayer stress (LBS) at the ER, the UPR triggers pathways to restore ER homeostasis and eventually induces apoptosis if the stress remains unresolved (*Celik et al., 2023*).

 

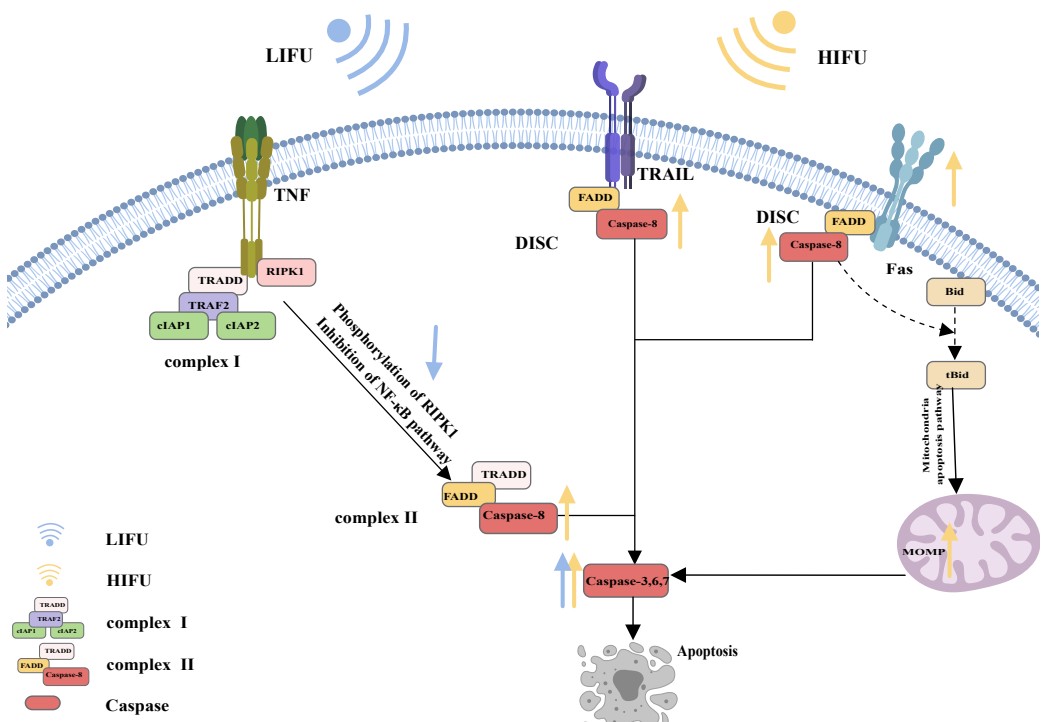

**Figure 1** **The death receptor pathway and the role of FUS in this pathway.** The figure depicts the apoptosis pathway and the changes in apoptosis-related protein expression in response to HIFU and LIFU. Created with MedPeer (www.medpeer.cn).

As a storage site for $Ca^{2+}$, the ER membrane can modulate its own luminal $Ca^{2+}$ dynamics and generate appropriate signals to maintain homeostasis, but disturbed $Ca^{2+}$ homeostasis activates ERS. During ERS, glucose-regulated protein 78 (GRP78), an especially important $Ca^{2+}$-binding protein, increases. RNA-activated protein kinase-like ER kinase (PERK) can dissociate from GRP78 and trigger autophosphorylation and oligomerization, activating eukaryotic initiation factor 2α (e IF2α). Activated eIF2α mediates the transcription of activated transcription factor 4 (ATF4), which induces the expression of the homologous protein (CHOP) (*Chen et al., 2022*; *Jeong et al., 2023*; *Mi et al., 2023*). In turn, CHOP induces increases in the expression of several proapoptotic proteins (Bak and Bax), enhances suppression of antiapoptotic proteins (Bcl-2 and Bcl-xl) and translocation of these proteins from the cytoplasm to the mitochondria, increases the concentration of $Ca^{2+}$ in the cytoplasm, and activates cysteine aspartate protease 12 (Caspase-12) and the caspase cascade (*Celik et al., 2023*; *Li et al., 2024*) (Fig. 3).

## Clearance of apoptotic cells

Under physiological conditions, phagocytes can engulf and clear apoptotic bodies in a timely manner to maintain homeostasis of the internal environment (this process is called efferocytosis; *Nagata & Segawa, 2021*). Macrophages, dendritic cells, epithelial cells, endothelial cells, and fibroblasts can all play a role in efferocytosis, with macrophages

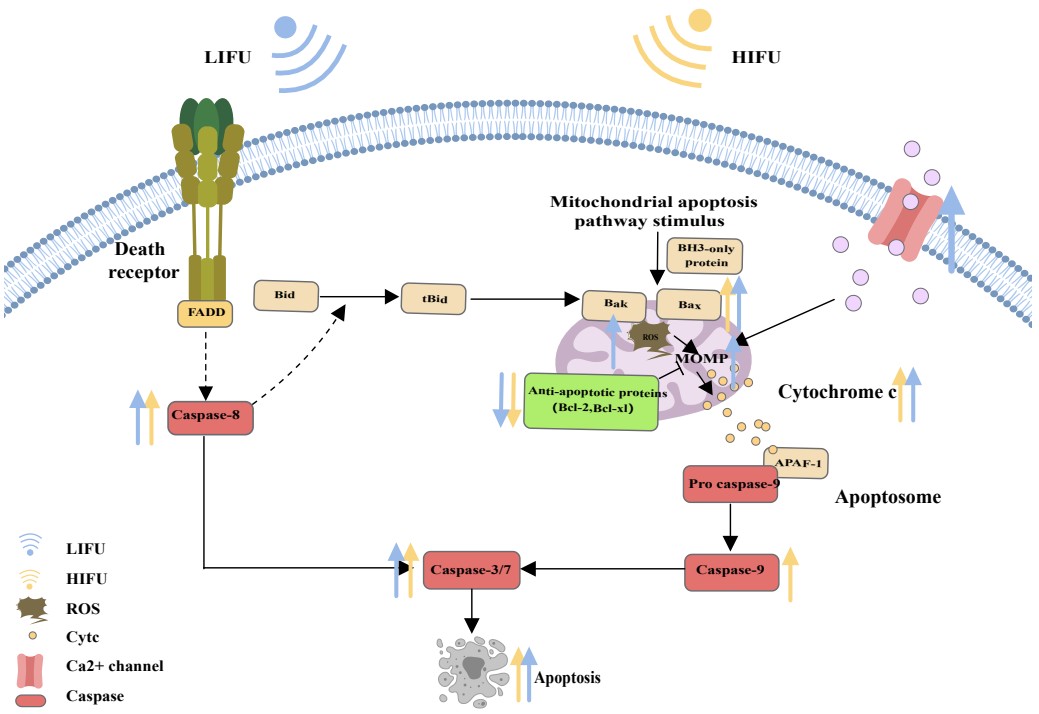

**Figure 2 The mitochondrial pathway and the role of FUS in this pathway.** The figure depicts the mitochondrial pathway of apoptosis and the changes in apoptosis-related protein expression in response to HIFU and LIFU. Created with MedPeer (www.medpeer.cn).

playing a major role (*Boada-Romero et al., 2020*). When cells undergo apoptosis, caspase-3 cleaves and inactivates flippases (ATP11A and 11C) and cleaves XKR8 to activate its phospholipid scramblase activity. Thus, phosphatidylserine (PtdSer) is rapidly and irreversibly exposed to the cell surface as an engulfment signal (*Nagata & Segawa, 2021*; *Ramos & Oehler, 2024*). PtdSer is recognized by macrophage receptors, including CD300b, BAI1, TIM4, and Stabilin-2. PtdSer is also recognized by soluble, bifunctional 'bridging' proteins, including Gas6/Pros1 and MFG-E8. The macrophage receptors for Gas6/Pros1 are the TAM receptor tyrosine kinases Mer and Axl, while those for MFG-E8 are αvβ3 and αvβ5 integrin dimers. Other proteins, including C1q, C3b and C4, also adhere to the surfaces of apoptotic cells by mechanisms that remain under study (*Lemke, 2019*). Macrophages recognize PtdSer and adhere to apoptotic cells.

Subsequently, apoptotic cells are usually engulfed by phagocyte lamellipodia, where Rac1 is activated. Activated Rac1 promotes actin polymerization and cytoskeletal rearrangement and the phagocytosis of apoptotic cells (*Segawa & Nagata, 2015*; *Bartneck et al., 2016*; *Henson, 2017*; *Nagata, 2018*) (Fig. 4).

### The role of FUS in apoptosis

Based on the process of apoptosis, this study explored whether HIFU and LIFU can promote apoptosis and the underlying mechanisms. HIFU (10 W/cm$^2$, 0.6 s, 1,584 kHz) can affect various aspects of apoptosis, including caspase-8 (a key participant in the death

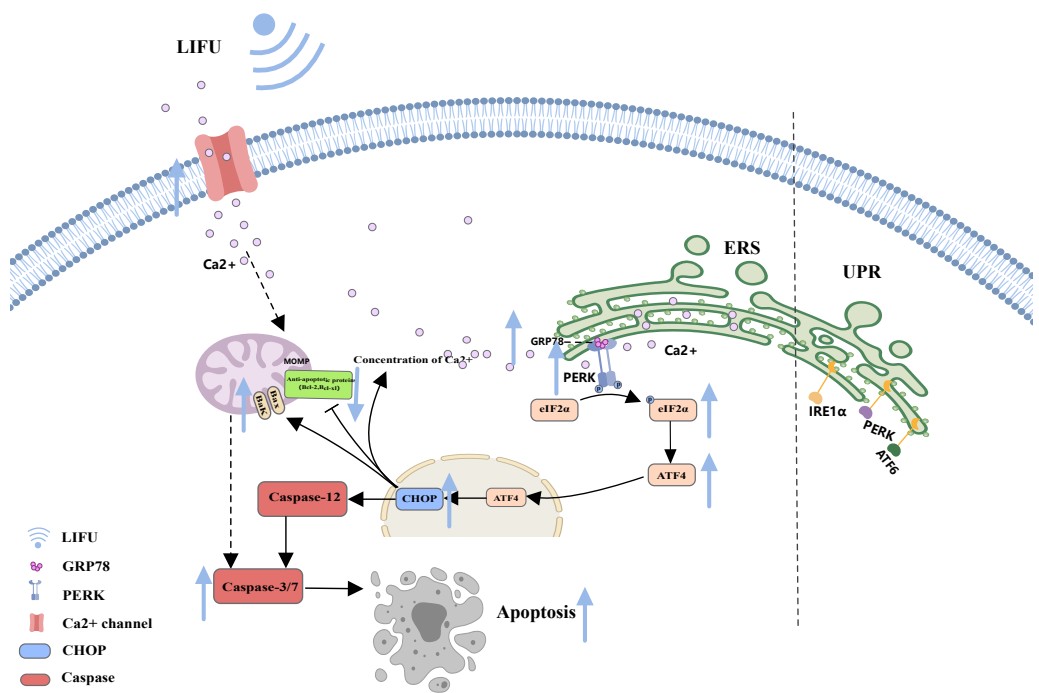

**Figure 3** **The endoplasmic reticulum stress pathway and the role of FUS in this pathway.** The figure depicts the apoptosis-associated ERS pathway and the changes in apoptosis-related protein expression in response to LIFU. Created with MedPeer (www.medpeer.cn).

receptor pathway), caspase-9 (a key participant in the mitochondrial apoptosis pathway), and caspase-3/6 (an apoptotic executor) (*Saliev et al., 2013*). *Ran et al. (2023)* reported that HIFU (5 W/cm$^2$, 160 s, 9.5 MHz) treatment increases the frequency of FasL, which can significantly induce apoptosis. *Zhong et al. (2019)* revealed the consequential production of ROS upon simultaneous HIFU (125 W/cm$^2$, 5 s, 0.94 MHz) irradiation. When ROS production increases, the mitochondrial membrane potential decreases, which causes mitochondrial apoptosis. *Byun et al. (2023)* reported that HIFU (0.6 J energy, 7 MHz) increased p53 translocation into mitochondria, which binds to Bcl-2 or Bcl-xl, causing Bak and Bax to be released from Bcl-2 or Bcl-xl. Thus, HIFU led to decreased expression of Bcl-2/Bcl-xl (an antiapoptotic signal) and increased expression of Bak/Bax (an apoptotic signal). *Zhang et al. (2017)* reported that HIFU (1,000 W/cm$^2$, 9 s, 1.048 MHz) exposure increased the expression of cleaved caspase-3 and PARP (a sign of apoptosis). *Fu et al. (2020)* reported that Fas expression was significantly upregulated after HIFU (3.5–4.5 W/cm$^2$, 10 MHz) treatment. To summarize, HIFU increases the expression of several apoptotic factors in the death receptor pathway (increasing FasL and caspase-8 expression) and the mitochondrial pathway (increasing caspase-9, Bax, Bak, and MOMP). HIFU also increases the expression of apoptotic executors (caspases −3, −6 and −7). However, this review did not find evidence that HIFU can affect the ERS pathway, which may necessitate further exploration and discussion in the future.

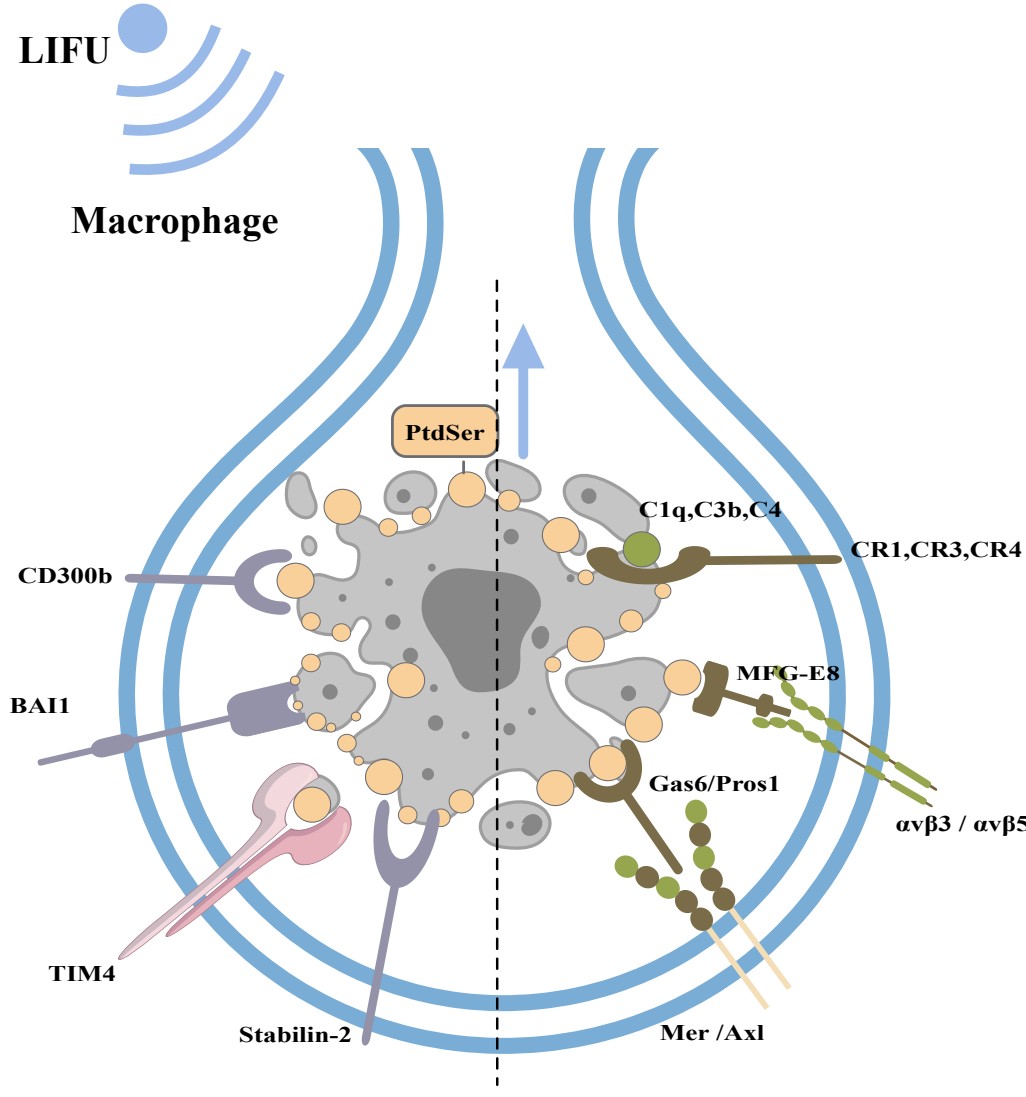

**Figure 4 Clearance of apoptotic cells and the role of FUS in this process.** The figure depicts apoptotic cell clearance and the types of apoptotic receptors. PtdSer is recognized by macrophage receptors, including CD300b, BAI1, TIM4, and Stabilin-2. PtdSer is also recognized by soluble, bifunctional 'bridging' proteins, including Gas6/Pros1 and MFG-E8. The macrophage receptors for Gas6/Pros1 are the TAM receptor tyrosine kinases Mer and Axl, while those for MFG-E8 are $\alpha$v $\beta$3 and $\alpha$v $\beta$5 integrin dimers. Created with MedPeer (www.medpeer.cn).

Similarly, this manuscript also reviews LIFU-induced apoptosis. *Yao et al. (2021)* reported that LIFU (1.5 W/cm², 90 s, 1 MHz) promotes mitochondrial-caspase apoptosis *via* ROS production, cytochrome C release and increased caspase-3 protein levels. Enhanced intracellular ROS levels are positively correlated with mitochondrial apoptosis (*Kuo et al., 2023*). *Hou et al. (2021)* suggested that Ca²⁺ endocytosis signalling occurs upon LIFU (0.2 MPa, 1 MHz) through activation of mechanosensitive ion channels. *Tabuchi et al. (2008)* observed the effects of LIFU (0.3 W/cm², 1 min, 1 MHz) on apoptosis, and PtdSer

externalization was examined using an annexin V-FITC kit. *Fang et al. (2023)* showed that an increase in LIFU (83.4 mW/cm$^2$, 25 min, 2 MHz) promoted the expression of the apoptosis marker proteins cleaved caspase-3 and p53 and increased the ratio of Bax. *Guo et al. (2023)* revealed that LIFU (1.6 W/cm$^2$, 60 s, 1 MHz) increased the mRNA expression levels of key apoptosis markers (Bad, Bax, caspase-9, and caspase-3), ultimately leading to cell apoptosis. LIFU (0.2 W/cm$^2$, 60 S, 360 kHz and 90 mW/cm$^2$, 200 Ms, 1.5 MHz) may promote apoptosis by inhibiting the NF-κB signalling pathway (*Liu et al., 2020b*; *Qiu et al., 2020*). *Liu et al. (2020b)* reported that LIFU (1 W/cm$^2$, 3 min, 1 MHz) could increase the expression of GRP78, PERK, CHOP and Bax and downregulate the expression of Bcl-2, triggering ERS-associated apoptosis and the mitochondrial apoptosis pathway. However, depending on the type of cancer cells and ultrasound parameters, the effects of HIFU and LIFU on the apoptotic pathway differ, as summarized in Table 1.

In conclusion, FUS can promote apoptosis by affecting apoptosis pathways, including the death receptor pathway, the mitochondrial apoptosis pathway, the ERS signalling pathway, and the clearance of apoptotic cells (*Shi et al., 2016*; *Ye et al., 2016*; *Jahagirdar et al., 2018*). If this effect can be controlled and utilized, it may facilitate biological response evaluations of the treatment area and reduce the incidence of complications when using FUS to treat high-risk cancer sites.

## Biological effects of FUS on apoptosis

The biological effects of FUS mainly include thermal effects, cavitation effects, mechanical effects, and molecular biological effects (*Zhang et al., 2020*; *Hu et al., 2023*). FUS can induce a series of biological reactions at the cellular and molecular levels. It can regulate the expression of different molecules or affect related signalling pathways to modulate malignant cell apoptosis (*Saliev et al., 2013*). This characteristic indicates that FUS is a promising adjuvant therapy for malignant tumours in the future. Moreover, HIFU locally heats and destroys diseased or damaged tissue through ablation (*Ashar & Ranjan, 2023*). In contrast, LIFU may have effects on cellular redox mechanisms, leading to the activation of heat shock proteins, dysregulation of cellular metabolic pathways, and apoptosis (*Li et al., 2022b*). The following section focuses on exploring the biological effects of LIFU on apoptosis.

### The role of thermal effects on apoptosis

The ultrasonic thermal effect refers to when LIFU propagates in biological tissues, tissues, and organs that can absorb ultrasonic energy, resulting in an increased regional tissue temperature (*Duan et al., 2020*). When the tissue inside focal area is heated and reaches temperatures near the thermal threshold, tumour cell apoptosis can be induced. The specific mechanism may be related to the production of heat shock protein (Hsp) and the induction of ischaemia and hypoxia through thermal effects (*Peng et al., 2019*). For example, Hsp90 can induce apoptosis *via* the death receptor pathway by stimulating RIPK1 expression and inhibiting the NF-κB pathway. Hsp90 can also lead to mitochondrial dysfunction and severe oxidative stress (*Gümüş et al., 2023*), including an increase in Apaf-1, which participates in the assembly of apoptotic bodies and activates caspase-3,

**Table 1 Effects on apoptosis induced by different FUS intensities.**

| | | Ultrasound parameters | Type of cancer cell line/-model | Apoptosis factors | Reference |
|---|---|---|---|---|---|
| **HIFU** | **Cell trials** | 10 W/cm$^2$, 0.6 s, 1584 kHz | KDH-8 cell lines | Upregulation of the expression of caspase-8, 9, 3, and 6 | *Saliev et al. (2013)* |
| | **Animal trials** | 5 W/cm$^2$, 160 s, 9.5 MHz | H22 tumor-bearing mice | Upregulation of the expression of FasL | *Ran et al. (2023)* |
| | | 125 W/cm$^2$, 5 s, 0.94 MHz | MDA-MB-231 tumor-bearing mice | Generation of large amounts of ROS and MOMP | *Zhong et al. (2019)* |
| | | 0.6 J energy, 7 MHz | Sprague-Dawley rats(high-fat diet) | Upregulation of the expression of BAX and BAK-Downregulation of the expression of Bcl-2 and Bcl-xl | *Byun et al. (2023)* |
| | | 1,000 W/cm$^2$, 9 s, 1.048 MHz | A594 tumor-bearing nude mice | Upregulation of the expression of caspase-3 and PRAP | *Zhang et al. (2017)* |
| | **Clinical trials** | 3.5–4.5 W/cm$^2$, 10 MHz | Cervical intraepithelial neoplasia 1 (CIN1) | Upregulation of the expression of Fas | *Fu et al. (2020)* |
| **LIFU** | **Cell trials** | 0.3 W/cm$^2$, 1 min, 1 MHz | U937 cell lines | Induction of PtdSer externalization | *Tabuchi et al. (2008)* |
| | | 1.6 W/cm$^2$, 60 s, 1 MHz | HepG2 cell lines | Upregulation of the mRNA expression levels of key apoptosis markers (Bad, Bax, caspase-9 and caspase-3) | *Guo et al. (2023)* |
| | | 90 mW/cm$^2$, 200 ms, 1.5 MHz | hPDLC cell lines | Inhibition of the NF-κB signalling pathway | *Liu et al. (2020b)* and *Liu et al. (2020a)* |
| | | 1 W/cm$^2$, 3 min, 1 MHz | SAS cell lines | Upregulation of the expression of GRP78, PERK, CHOP and BAX | *Liu et al. (2020b)* and *Liu et al. (2020a)* |
| | **Animal trials** | 0.2 W/cm$^2$, 60 s, 360 kHz | ASPC-1 tumor-bearing nude mice | Inhibition of the NF-κB signalling pathway | *Qiu et al. (2020)* |
| | | 1.5 W/cm$^2$, 90 s, 1 MHz | Adult male New Zealand rabbits (high-cholesterol diet) | ROS induction, cytochrome c release, caspase 3 upregulation | *Yao et al. (2021)* |
| | | 0.2 MPa, 1 MHz | Mouse brains with GVs injected in a specific region. | Ca$^{2+}$ endocytosis | *Hou et al. (2021)* |

caspase-6, and caspase-9 (*Wu et al., 2019*; *Hu et al., 2020a*; *Peng et al., 2022*). On the other hand, ischaemia and hypoxia lead to an apoptotic response in the ERS pathway through increased expression of the proapoptotic transcription factors CHOP and GRP78 (*Liu et al., 2020a*).

In summary, the introduction of LIFU close to the threshold can stimulate the expansion or enhancement of apoptosis and protect surrounding normal tissues and cells, representing a reliable method for improving the safety and efficacy of FUS in tumour treatment.

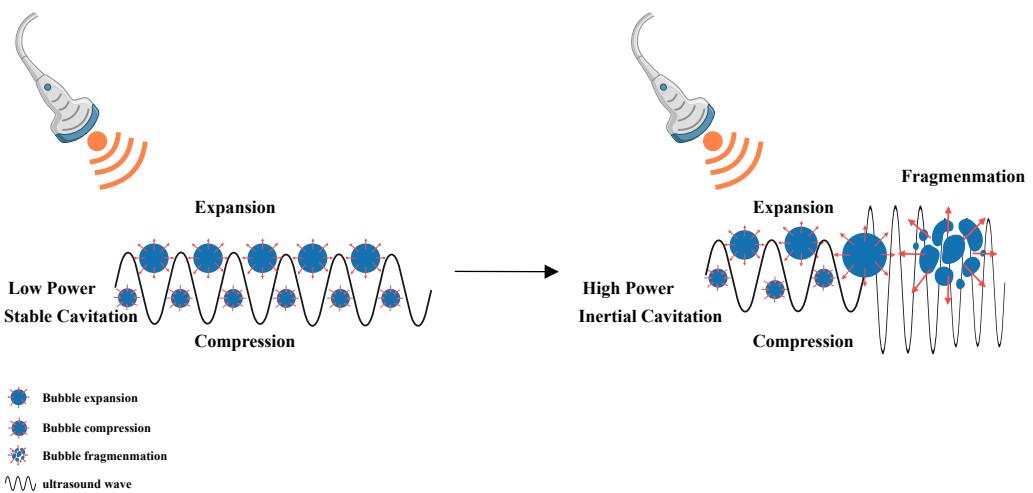

**Figure 5 Cavitation induced by LIFU.** This figure depicts the cavitation effect produced by LIFU. When the LIFU energy is sufficient, the tiny bubbles present in the liquid vibrate, grow and accumulate ultrasonic energy. When the energy reaches a certain threshold, the cavitation bubbles collapse and rapidly rupture, which is the mechanism underlying the cavitation effect of ultrasound. Created with MedPeer (www.medpeer.cn).

### The role of the cavitation effect in apoptosis

When the LIFU energy is sufficient, the microbubbles present in the liquid vibrate, grow and accumulate ultrasonic energy. When the energy reaches a certain threshold, the cavitation bubbles collapse and rapidly rupture, which is the mechanism underlying the cavitation effect of ultrasound (*Myers et al., 2018*; *Padilla et al., 2023*) (Fig. 5). Introduction of the echo contrast agent Levovist and targeting of malignant cells with therapeutic LIFU cause transient cavitation in the focal area, which results in significant cell apoptosis-related morphological changes, including cell contraction, membrane blebbing, chromatin aggregation, nuclear fragmentation, and the formation of apoptotic bodies (*Ando et al., 2006*; *Myers et al., 2018*). LIFU has a significant dose-dependent effect on cell apoptosis within a certain range of radiation doses, with an obvious increase in cell apoptosis as the radiation dose increases (*Shi et al., 2016*; *Hu et al., 2023*).

This part of the present review explored the pathway through which the LIFU-mediated cavitation effect induces apoptosis. *Cao et al. (2021)* confirmed that LIFU increased cleaved caspase-3 levels, decreased Bcl-2 levels, increased intracellular $Ca^{2+}$ concentrations and increased cleaved caspase-8 (1.4 $W/cm^2$, 1 min, 50% duty cycle, 360 kHz)-mediated cavitation. *Zhao et al. (2015)* observed that LIFU (1 MHz, 0.3-MPa peak negative pressure, 10% duty cycle, and 1-kHz pulse repetition frequency)-assisted cavitation can increase the cellular apoptotic index, mitochondrial depolarization, and cytochrome c release. LIFU (2 $W/cm^2$, 40% duty cycle, 20 kHz) can induce mitochondrial depolarization, inner MOMP, and mitochondria-caspase signalling pathway activation together with increased $Ca^{2+}$ concentrations and different expression levels of caspase-3, Bcl-2, and Bax, thus inducing the apoptosis of carcinoma cells (*Shen et al., 2020*). *Ho et al. (2023)* reported that

**Table 2  Effects on apoptosis induced by LIFU-mediated cavitation.**

| | Ultrasound parameters | Type of cancer cell line/-model | Apoptosis factors | Reference |
|---|---|---|---|---|
| **Cell trials** | 1.4 W/cm$^2$, 1 min, 360 kHz, 50% duty cycle | AsPC-1 cells | Caspase- 3, caspase-8, intracellular Ca$^{2+}$,Bcl-2 | *Cao et al. (2021)* |
| | 0.3 MP, PRF 1 Hz, 10% duty cycle, 1 MHz | K562 cells | MOMP, cytochrome c release | *Zhao et al. (2015)* |
| | 2.5 W/cm$^2$ | HEp-2 cells | Caveolin-1, STAT3 signalling pathway | *Ye et al. (2016)* |
| | 0.5 MHz, 210 mW/cm$^2$ | HUVEC cells | p38-mediated MAPK pathway, ATF- 4, eIF2 $\alpha$ | *Su et al. (2019)* |
| **Animal trials** | 1 MHz, PRF 1 Hz, 300 kPa | Ischaemia-stroke reperfusion model | NF-κB, Bcl-2 | *Ho et al. (2023)* |
| | 1 MHz, 1 w/cm$^2$, 50% duty cycle | Psoriasis-like mouse model | ROSMitochondrial dysfunction | *Xi et al. (2022)* |

cavitation induced by LIFU (1 MHz, 5,000 cycles, 1 Hz, 300 kPa) can inhibit the NF-κB signalling pathway. After stimulation with LIFU (1 MHz, 1 W/cm$^2$, 50% duty cycle), cavitation generated abundant intracellular ROS, which caused HaCat cell apoptosis by inducing mitochondrial dysfunction (*Xi et al., 2022*). *Ye et al. (2016)* discovered that LIFU (2.5 W/cm$^2$) can also downregulate the expression of the caveolin-1 protein and inhibit the STAT3 signalling pathway, hindering normal cell growth and redirecting apoptosis. *Su et al. (2019)* demonstrated that LIFU (210 mW/cm$^2$, 1 min, 20.5 MHz) can induce the p38-mediated MAPK and ERS apoptosis pathways, including ATF-4 and eIF2α activation (Table 2).

LIFU-induced cavitation may participate in the mitochondrial pathway, death receptor pathway, and ERS pathway. LIFU is a novel apoptotic approach for tumour therapy, and key apoptotic factors, such as Bcl-2, caspase-8, and caspase-9, also warrant increased attention. For example, *Pan et al. (2022)* introduced BH3 mimetics (specific inhibitors of Bcl-2 and Bcl-xl) to increase mitochondrial sensitization in cancer cells and directly induce apoptosis. In the future, we look forward to more studies focusing on this topic.

### LIFU enhances the clearance of apoptotic cells by macrophages

Macrophages are the main contributors to apoptotic cell elimination. At the early stage of tumour cell apoptosis, various structural changes occur on the cell membrane surface, such as PtdSer externalization. Corresponding receptors on the surface of the macrophage membrane can recognize the above substances, adhere to apoptotic cells, and promote phagocytosis (*Le et al., 2024*). The subtypes of macrophages mainly include M1 and M2 (*Peng et al., 2023*). M1 macrophages can kill tumour cells and inhibit tumour growth through phagocytosis and the Th1 response. M2 macrophages promote tissue repair, angiogenesis, and immune suppression by producing cytokines and triggering a Th2 response, which further facilitates tumour progression (*Peng et al., 2023*; *Wang et al., 2023a*). *Kong et al. (2021)* reported that LIFU (90 W/cm$^2$*10 min, 80 kHz) can cause vibration of the piezoelectric material β-PVDF and local charge release, which opens voltage-gated channels to Ca$^{2+}$ influx and stimulates macrophage M1 polarization and

M2 macrophages to transform into M1 macrophages through the $Ca^{2+}$-CAMK2A-NF-κB signalling pathway. While engulfing apoptotic cells, M1 macrophages secrete a large number of proinflammatory cytokines, inhibit the activity of cocultured tumour cells, and kill tumour cells (*Kong et al., 2021*). Therefore, under LIFU stimulation, the apoptosis of malignant cells increased, and the clearance of apoptotic cells by macrophages also increased. This synergistic effect may enhance the therapeutic effect of LIFU on malignant tumours.

## Combining FUS to promote tumour treatment *via* cell apoptosis

Currently, the use of FUS in clinical practice mainly focuses on the thermal ablation effect caused by HIFU, aiming for thermal coagulative necrosis. However, FUS-induced apoptosis has advantages in terms of medium- to long-term effects and complication control in tumour treatment. FUS is often combined with microbubbles, drugs, gene transfection, and other modalities to enhance the induction of tumour cell apoptosis and improve the effectiveness of tumour treatment. In the following section, we discuss the application of FUS-induced apoptosis combined with other treatment modalities in tumours. Depending on the specific cancer cell type, the ultrasound parameters used may also vary. We have summarized the information in Table 3.

### FUS combined with microbubbles to induce apoptosis

Microbubbles can not only serve as ultrasound contrast agents but also be oscillated and activated by FUS, enhancing the inherent biological effects of ultrasound, which facilitates the destruction of tumour cells while reducing damage to surrounding normal tissues and improving clinical safety (*Huang et al., 2023*). FUS can induce multiple stimulating effects to promote cell apoptosis, such as reducing the mitochondrial membrane potential, promoting oxidative stress, and inducing extracellular $Ca^{2+}$ influx and cell skeleton rearrangement. The introduction of microbubbles can decrease the threshold for achieving biological effects with FUS, thus facilitating the induction of apoptosis (*Przystupski & Ussowicz, 2022*).

Sondynamic therapy (SDT), which involves deep tissue penetration and high precision, has strong potential to become an ideal tumour therapy. This section focuses on the effects of SDT on apoptosis. Under FUS irradiation, sonosensitizers can be activated from the ground state to the excited state to generate ROS, including singlet oxygen and hydroxyl radicals. Acoustic cavitation plays an important role in activating sonosensitizers to generate ROS during the interaction of FUS with an aqueous environment (*Zeng et al., 2022*; *Zhang et al., 2022*). $Ca^{2+}$ overload plays a primary role in the apoptotic process and is associated with increased ROS production, decreased mitochondrial membrane potential, and increased cyt-c (*Bunevicius et al., 2022*). In addition, SDT induces ER stress, Bcl-2 downregulation, Bax upregulation, and ultimately tumour cell apoptosis (*Wang et al., 2024*). Currently, studies on the use of SDT to induce apoptosis for treatment have involved several cancer types, such as glioblastoma, pancreatic cancer, squamous cell carcinoma, spinal-metastasized tumours, triple-negative breast cancer, osteosarcoma, prostate cancer, and hepatocellular carcinoma (*Aksel et al., 2021*; *Wang et al., 2022*; *Wang et al., 2023a*;

**Table 3 Apoptosis induced by FUS combined with microbubbles.**

| | | | Ultrasound parameters | Type of cancer cell line/model | Mechanisms | Reference |
|---|---|---|---|---|---|---|
| FUS-sensitizing radiotherapy | Cell trials | | 1,136 W/cm$^2$, 40 s | FaDu, T98G, PC-3 cells | FUS-induced cavitation | *Hu et al. (2020a)* and *Hu et al. (2020b)* |
| | | | 225 W/cm$^2$, 1.467 MHz | FaDu, T98G, PC-3 cells | DNA damage | *Zhang et al. (2021)* |
| | Animal trials | | 3.5 W/cm$^2$, 1 MHz | 4T1 tumor-bearing mice | Oxygenation enhancement | *Xiao et al. (2024)* |
| | | | 740 kPa, 5 min, 500 kHz | MDA-MB-231 tumor-bearing mice | Cell death, microvascular effects | *Sharma et al. (2023)* |
| FUS-sensitizing chemotherapy | Animal trials | | 10-ms pulse length, 1-Hz pulse repetition frequency, 0.64-MPa peak-rarefactional pressure | nude mice bearing intracranial glioblastoma | Enhanced delivery of paclitaxel liposomes | *Shen et al. (2017)* |
| | | | 24-fold transmission in 1 cycle, 3.5 MHz | 4T1 tumor-bearing BAL-B/c mice | Sonosensitized electron transfer, ROS-mediated mitochondrial damage, cell apoptosis | *Liu et al. (2023)* |
| Microbubble-mediated gene transfection | Cell trials | | 0.5 W/cm$^2$, 8 s, 1 MHz | OVCA-433 cells | Caspase-3, caspase-8 | *Xu et al. (2021)* |
| | | | 1.2 W/cm$^2$, 20 s, 20% duty cycle | VCaP, LNCaP, PC-3, DU145 cells | Upregulation of the protein expression levels of the apoptosis-associated genes caspase-9, cleaved caspase 9, cytochrome c | *Qin, Li & Xie (2018)* |
| | Animal trials | | 21-MHz (MS250, Visual Sonics, Toronto, Ontario, Canada) | HepG2, HepG3 tumor-bearing nude mice | Inhibition of cancer cell proliferation at the gene level | *Chowdhury et al. (2018)* |
| | | | 1.2 MHz, 60 cycles, 5.5 MPa, 40-Hz pulse repetition frequency | SMMC-7721 tumor-bearing nude mice | Cavitation effect | *Dong et al. (2020)* |
| | | | 0.5 W/cm$^2$, 8 s, 1 MHz | OVCA-433 cells | Caspase-3, caspase-8 | *Xu et al. (2021)* |
| Microbubble-mediated gas transfection | Animal trials | NO | 1.5 W/cm$^2$, 5 min, 20% duty cycle, 1 MHz | 4T1 tumor-bearing mice | Interaction with ROS | *Chen et al. (2023)* |
| | | NO | 1.0 MHz, 50% duty cycle, 1.0 W/cm$^2$ | Lung metastasis bearing mice | Induction of strong intracellular oxidative stress levels and DNA double-strand breaks | *Wang et al. (2021b)* |
| | | H2S | 30 kHz, 5 min | pulmonary metastasis of 4T1 breast tumours | Inhibition of mitochondrial respiration and ATP generation | *Li et al. (2022a)* and *Li et al. (2022b)* |
| | | CO | 1 W/cm$^2$, 1 MHz, 10 s | 4T1 tumor-bearing mice | reducing mitochondrial membrane potential | *Guo et al. (2022)* |

*Tian et al., 2023*; *Gong et al., 2023*). Furthermore, related clinical trials (NCT05362409 and NCT05580328) have evaluated the antitumour efficacy of sonosensitizer therapy and indicates that SDT has favourable clinical potential.

## FUS combined with chemotherapy/gene/gas therapy induces apoptosis in tumour cells

*FUS enhances the ability of radiotherapy and chemotherapy to induce apoptosis in tumour cells.* FUS can sensitize cells to radiotherapy. The endothelial cell membrane is subjected to mechanical damage under ultrasound stimulation, rendering it more responsive to radiation therapy and enhancing the effect of the original treatment (*Shi et al., 2021*; *McCorkell et al., 2022*). *Leong et al. (2023)* demonstrated that the use of FUS (750 kPa, 500 kHz)-stimulated microbubbles can potentially enhance the effects of radiotherapy through the activation of the acid sphingomyelinase ASMase or sphingomyelin phosphodiesterase 1 (SMPD1)-ceramide pathway. The findings of *Xiao et al. (2024)* strongly support the role of nano-PFC as a US (3.5 W/cm$^2$, 1 MHz)-responsive oxygen carrier in improving the radiosensitizing effect by enhancing tumour oxygenation. *Sharma et al. (2023)* demonstrated that for MDA-MB-231 xenograft tumours, significant cell death occurs with ultrasound treatments as short as 1 min, and significant microvascular effects require a longer treatment time (>5 min). Using an *in vitro* cell culture model, *Hu et al. (2020b)* showed the potential of FUS (740 kPa, 5 min, 500 kHz)-induced cavitation as a sensitizer to radiotherapy. *Zhang et al. (2021)* reported that FUS (213/225 W/cm$^2$, 1.14/1.467 MHz) radiosensitizes human cancer cells by enhancing DNA damage. *Eisenbrey et al. (2021)* reported that the combination of FUS (2.3-μsec pulses at a pulse repetition frequency of 100 Hz, 1.13-μsec at 1.5 MHz)-triggered microbubble destruction and transarterial radioembolization is feasible with an excellent safety profile in their patient population and appears to result in an improved hepatocellular carcinoma treatment response.

Similarly, FUS can also sensitize cells to chemotherapy. *Shen et al. (2017)* used microbubbles to enhance the delivery of paclitaxel liposomes to treat intracranial glioblastoma. Microbubbles not only enhance blood–brain barrier penetration and increase drug accumulation in local tumour tissue but also enhance the proapoptotic effect of paclitaxel liposomes. Paclitaxel can exert unique antitumour effects, which include inhibiting cell division, suppressing cell proliferation, increasing chromosomal instability, and promoting apoptosis (*Scribano et al., 2021*). *Liu et al. (2023)* developed cyanin platin, a Pt(IV) prodrug that can be controllably activated by FUS (24-times transmission in 1 cycle is used to achieve a video-rate imaging speed (17 frames/s), 3.5 MHz). Upon irradiation with FUS, the prodrug was reduced to chemotherapeutic carboplatin *via* a sonosensitized electron transfer process. Simultaneously, sonoactivated cyaninplatin generated ROS and depleted intracellular reductants, thereby enhancing ROS-mediated mitochondrial damage and cell apoptosis efficiency. *Luo et al. (2022)* discovered that microbubbles can be used as sensitizers during chemotherapy. The optimal treatment time is immediately after chemotherapy. This approach can significantly increase drug perfusion and improve the effectiveness of killing tumour cells. Based on the characteristics of FUS and microbubbles, we can reduce the dosage of chemotherapy drugs or the radiation intensity.

*The combination of FUS and microbubble-mediated gene transfection induces apoptosis in tumour cells.* Researchers have also shown that UTMD transects small-molecule miRNAs

into cells at the gene level. miRNAs can upregulate the expression of proapoptotic proteins or downregulate the expression of proapoptotic proteins by integrating genetic information (*Chowdhury et al., 2018*; *Qin, Li & Xie, 2018*; *Ran et al., 2018*; *Michon, Rodier & Yu, 2022*). *Qin, Li & Xie (2018)* UTMD (1.2 W/cm$^2$, 20 s; 20% duty cycle) was used to transfect siRNA205 into prostate cells, which upregulated the protein expression levels of the apoptosis-associated genes caspase-9, cleaved caspase- 9, and cytochrome c and successfully inhibited the proliferation, migration, and invasion of prostate cancer cells. *Chowdhury et al. (2018)* delivered complementary miRNA122 into hepatocellular carcinoma through microbubble vectors, inhibited the proliferation of cancer cells at the gene level, and promoted their apoptosis. *Dong et al. (2020)* combined PLNDs with UTMD (1.2 MHz, 60 cycles, 5.5 MPa, 40-Hz pulse repetition frequency), delivered four pre-miRNA plasmids, and verified their therapeutic efficacy in subcutaneous tumours in a mouse xenograft HCC model. *Xu et al. (2021)* synthesized a targeted microbubble agent for UTMD (0.5 W/cm$^2$, 8 s, 1 MHz)-mediated shRNA against the Livin gene in human ovarian cancer OVCA-433 cells. Livin activity is associated with the expression of caspase-3 and caspase-8.

*The combination of FUS and gas therapy induces tumour cell apoptosis.* *Guo et al. (2022)* constructed an efficient ultrasonic-triggered and targeted CO release strategy based on a novel targeted acoustic release carrier of carbon monoxide (TARC-CO). With FUS (1 W/cm$^2$, 1 MHz, 10 s) irradiation, CO was demonstrated to effectively induce mitochondrial dysfunction by reducing the mitochondrial membrane potential, leading to the apoptosis of 4T1 cells. Moreover, different gas therapies, such as NO and H$_2$S, have been investigated extensively (*Chen et al., 2023*). *Chen et al. (2023)* produced a unique strategy for producing NO gas that was successfully developed *via* FUS (1.5 W/cm$^2$, 5 min, 20% duty cycle, 1 MHz)-induced piezo catalysis-based polyarginine-coated barium titanate nanoparticles (BTO@DPA). NO can even further interact with ROS to produce more reactive peroxynitrite, thus inducing serious tumour cell apoptosis under both hypoxia and normoxia. *Wang et al. (2021a)* reported that NO in cancer cells can causes strong intracellular oxidative stress and DNA double-strand breaks to ultimately induce cancer cell apoptosis. *Li et al. (2022a)* reported that H$_2$S-mediated inhibition of mitochondrial respiration and ATP generation promotes cell necrosis and apoptosis.

If microbubbles carry therapeutic gas into the local tumour site and release that gas under FUS energy, they can avoid damaging the surrounding normal tissues while rapidly achieving a suitable concentration of proapoptotic gas, serving as a potential means for future tumour-specific therapy.

## CONCLUSION

As a conventional medical tool, FUS is often used for the diagnosis and treatment of diseases. Relevant studies have recently shown that FUS can promote tumour cell apoptosis through three major mechanisms—thermal effects, cavitation effects, and related molecular biological effects—and therefore shows promise for minimally invasive treatment of tumours. Moreover, FUS combined with microbubbles can also enhance the clearance of apoptotic cells and enhance the targeting and sensitivity of tumour therapy. The specific

molecular mechanism by which FUS promotes apoptosis has not yet been elucidated, and more researchers need to focus on this topic in the future. Because the energy required to achieve FUS varies for different lesion sites, determining how to regulate the appropriate FUS energy and accurately define the irradiation target area to improve the effectiveness of tumour treatment while ensuring safety also needs to be prioritized in future studies.

### Funding
This work was supported by the Natural Science Foundation of Chongqing, China (No. cstc2020jcyj-msxmX0538, No. cstb2022nscq-msxc324) and the China Postdoctoral Science Foundation (2023M740439). The funders had no role in study design, data collection and analysis, decision to publish, or preparation of the manuscript.

### Grant Disclosures
The following grant information was disclosed by the authors:
The Natural Science Foundation of Chongqing, China: No. cstc2020jcyj-msxmX0538, No. cstb2022nscq-msxc324.
The China Postdoctoral Science Foundation: 2023M740439.

### Competing Interests
The authors declare that they have no conflict of interest.

### Author Contributions
- Na Wang conceived and designed the experiments, performed the experiments, analyzed the data, prepared figures and/or tables, authored or reviewed drafts of the article, and approved the final draft.
- Li Luo conceived and designed the experiments, performed the experiments, analyzed the data, authored or reviewed drafts of the article, and approved the final draft.
- Xinzhi Xu performed the experiments, authored or reviewed drafts of the article, and approved the final draft.
- Hang Zhou conceived and designed the experiments, authored or reviewed drafts of the article, and approved the final draft.
- Fang Li conceived and designed the experiments, authored or reviewed drafts of the article, and approved the final draft.

### Data Availability
This article is a literature review and did not utilize raw data.

### Supplemental Information
Supplemental information for this article can be found online at http://dx.doi.org/10.7717/peerj.17886#supplemental-information.

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
