# Peer review of "Focused ultrasound-induced cell apoptosis for the treatment of tumours"

_PeerJ, doi:10.7717/peerj.17886_

## Round 0.1 · original submission · Major Revisions

· Academic Editor

Major Revisions

This manuscript needs major revision for several key reasons:

Clarity and Structure
The manuscript lacks clarity due to issues like improper line spacing, incomplete explanations, and unstructured sentences. There's a need for better organization of thoughts and logical arguments supported by evidence.
The introduction is insufficient in providing the latest information and rationale for the literature survey.

Content Depth and Coverage
The manuscript lacks thorough explanations on critical aspects, such as how ultrasound accurately targets tumors, categorization of ultrasonic effects, and the feasibility of targeting molecules within the focal area.
There's a need to discuss newer references (up to 2023-2024) in each section and include a detailed survey methodology, possibly with a list of all keywords used.

Experimental Design and Validity
The discussion on sonodynamic therapy, low-intensity ultrasound, and targeting of apoptosis-related molecules requires more precision and updated information.

The manuscript needs to address safety concerns related to ultrasound intensity and provide clear definitions and intensity ranges.

Figures and Tables
The manuscript lacks adequate summary tables, graphs, or figures that would help clarify concepts related to focused ultrasound protocols and effects.
Existing figures and tables need proper formatting, legends, and acknowledgments.

Overall Review Quality
The manuscript should critically analyze the literature rather than narrate it, addressing extensive literature gaps, suboptimal English, and typological errors.
The abstract must reflect the actual content discussed, and the survey methodology needs clarification on whether it's a meta-analysis or literature review.

In summary, a major revision is essential to enhance the manuscript's clarity, depth, and overall quality. This involves restructuring for coherence, addressing specific technical gaps, updating references, and improving visual aids to support the discussion effectively by implementing suggestions made by reviewers.

**Language Note:** The review process has identified that the English language must be improved. PeerJ can provide language editing services - please contact us at [email protected] for pricing (be sure to provide your manuscript number and title). Alternatively, you should make your own arrangements to improve the language quality and provide details in your response letter. – PeerJ Staff

Reviewer 1 ·

Basic reporting

Attached

Experimental design

Attached

Validity of the findings

Attached

Additional comments

Attached

Annotated reviews are not available for download in order to protect the identity of reviewers who chose to remain anonymous.

Reviewer 2 ·

Basic reporting

The authors wrote a nice review. However, they must include newer references in each section, 2023 and 2024.
Maybe, for better clearance, in the Survey Methodology part, they shall include the timeframe of the references they have used and the keywords they have searched for.
The technical editor shall check the technical organization of the manuscript, reference style, etc...

It would be better if the paper had summary tables (papers, facts) and/or graphs explaining FUS, protocols....exp: https://www.sciencedirect.com/science/article/abs/pii/S0168365923001037 , https://www.sciencedirect.com/science/article/abs/pii/S0939641124000729

Experimental design

Need to update references to 2023 and 2024

Validity of the findings

no comment'

Reviewer 3 ·

Basic reporting

The authors provide a report on extensive research of literature on the mechanisms of FUS induced apoptosis.
The paper is hard to read because the line spacing has not been respected at large.
In the introduction the authors list cancer treatments that are globally used but only name a few and then cut short with a “and so on”, Line 28-29. It is not clear why the other therapy forms are cut short. Restructuring of the sentence with justification why the explicitly named therapy forms are the most important ones.
In lines 33-34 the authors state that ultrasound can accurately target tumor regions. However, throughout the manuscript the authors never mention how the accurate targeting is done.
In line 34- 35 the authors list the effects which ultrasound can achieve to induce tumor cell apoptosis, however they list over specific effects and then cut short with a “and so on”. They should better categorize the ultrasonic effects in mechanical effects, thermal effects, non-thermal and non-mechanical effects.
In lines 63-64 the authors don’t make clear why programmed cell death is feasible.
In lines 191-195 it is not convincingly explained why near-threshold ultrasound intensity stimulates apoptosis in tumors but won’t in the surrounding tissue.

Statistical graphics and/or a table on clinical trials sorted by FUS treatment modality would provide the reader with a better overview of the current state of the art.

Experimental design

While reactive oxygen species are mentioned, sonodynamic therapy is not been taken into consideration, which is a good candidate for the treatment of glioblastoma (First Participant Receives Sonodynamic Therapy for Glioblastoma in New Clinical Trial - Focused Ultrasound Foundation (fusfoundation.org), https://doi.org/10.1007/s11060-021-03807-6).
Description of the Survey Methodology from line 48-54 only shows a brief selection of the used keywords. The authors should include a list of all used keywords in the appendix. Furthermore, the authors should include the number of analysed articles and the number of articles included in the review.
In the introduction Line 29-31 it should the source for the five-year survival rates should be referenced.

Validity of the findings

In line 38 the authors imply to target apoptosis-related molecules. However, the focal are of a FUS acoustic field extends in the order of mm, whereas molecules usually have a diameter of several Å – nm, and targeting premises seeing the target somehow. Therefore targeting of a molecule is unrealistic. More suitable was sonication of apoptosis-related molecules at larger areas would describe the aim more accurately. The same holds for line 244-245, where the authors state that the mitochondria is targeted.
In line 113-118 the authors state that low intensity ultrasound can promote apoptosis. However this statement results in general safety concerns for the use of ultrasound. Neither diagnostic ultrasound nor usage of low intensity focused ultrasound is known to have apoptosis as a side effect. The authors should give a definition of the Intensity range they assign for low intensity ultrasound.

The main critical issue is that the paper lacks from a thorough discussion. The foremost aspect was how to prove apoptosis versus cell ablation with destruction of proteins and cell structures. This can only been proven by histomorphological examination. This has not been covered as a major review aspect.

Additional comments

The paper requires substantial major revision

---

## Round 0.2 · Minor Revisions

· Academic Editor

Minor Revisions

Please include MR-guided targeting of tumor regions in lines 33-34.
Include differentiation between cell experiments, animal trials and clinical trials in the tables.

Histomorphological examination as a means to prove apoptosis versus cell ablation with destruction of proteins and cell structures are essential and should be included.

Reviewer 2 ·

Basic reporting

The authors have implemented all the reviewer's requests.

Experimental design

The authors have implemented all the reviewer's requests.

Validity of the findings

The authors have implemented all the reviewer's requests.

Additional comments

The authors have implemented all the reviewer's requests.

Reviewer 3 ·

Basic reporting

We appreciate that the authors discussed our criticism and clarified how targeting of tumor regions is achieved. However, the authors lack mentioning MR-guided targeting of tumor regions in lines 33-34.
The Authors included tables which provides the reader with a better overview of the methods of using FUS in combination with other approaches to induce apoptosis for cancer treatment found in literature. These tables are helpful for the reader but the content is misleading. Therfore we strongly suggest the differentiation between cell experiments, animal trials and clinical trials. Especially listing of the clinical trials would be important to provide the reader an overview of the current advances in the field.
Line 242 is misleading, as it suggests that a focus is exposed. However, it is the tissue inside focal area that is heated and reaches temperatures near the thermal threshold.

Experimental design

see below

Validity of the findings

Histomorphological examination as a means to prove apoptosis versus cell ablation with destruction of proteins and cell structures are essential to used and be discussed. This is unfortunately still missing in the manuscript and must be included.

Additional comments

The revised version of the manuscript, shows good improvement in overall quality. The paper has still a few shortcomings:

---

## Round 0.3 · accepted · Accept

· Academic Editor

Accept

Your manuscript can be accepted.